# Theranostic Properties of Crystalline Aluminum Phthalocyanine Nanoparticles as a Photosensitizer

**DOI:** 10.3390/pharmaceutics14102122

**Published:** 2022-10-06

**Authors:** Vladimir I. Makarov, Daria V. Pominova, Anastasiya V. Ryabova, Igor D. Romanishkin, Arina V. Voitova, Rudolf W. Steiner, Victor B. Loschenov

**Affiliations:** 1Prokhorov General Physics Institute, Russian Academy of Sciences, Moscow 119991, Russia; 2Department of Laser Micro-, Nano- and Biotechnologies, National Research Nuclear University MEPhI, Moscow 115409, Russia; 3Faculty of Chemistry, Lomonosov Moscow State University, Moscow 119991, Russia; 4Institute for Laser Technologies in Medicine and Metrology (ILM) at Ulm University, 89081 Ulm, Germany

**Keywords:** nanoparticles, phototheranostic, photosensitizer, aluminum phthalocyanine, fluorescence lifetime, phototoxicity

## Abstract

The study of phthalocyanines, known photosensitizers, for biomedical applications has been of high research interest for several decades. Of specific interest, nanophotosensitizers are crystalline aluminum phthalocyanine nanoparticles (AlPc NPs). In crystalline form, they are water-insoluble and atoxic, but upon contact with tumors, immune cells, or pathogenic microflora, they change their spectroscopic properties (acquire the ability to fluoresce and become phototoxic), which makes them upcoming agents for selective phototheranostics. Aqueous colloids of crystalline AlPc NPs with a hydrodynamic size of 104 ± 54 nm were obtained using ultrasonic dispersal and centrifugation. Intracellular accumulation and localization of AlPc were studied on HeLa and THP-1 cell cultures and macrophages (M0, M1, M2) by fluorescence microscopy. Crystallinity was assessed by XRD spectroscopy. Time-resolved spectroscopy was used to obtain characteristic fluorescence kinetics of AlPc NPs upon interaction with cell cultures. The photodynamic efficiency and fluorescence quantum yield of AlPc NPs in HeLa and THP-1 cells were evaluated. After entering the cells, AlPc NPs localized in lysosomes and fluorescence corresponding to individual AlPc molecules were observed, as well as destruction of lysosomes and a rapid decrease in fluorescence intensity during photodynamic action. The photodynamic efficiency of AlPc NPs in THP-1 cells was almost 1.8-fold that of the molecular form of AlPc (Photosens). A new mechanism for the occurrence of fluorescence and phototoxicity of AlPc NPs in interaction with cells is proposed.

## 1. Introduction

The study of phthalocyanines (Pcs), known photosensitizers (PSs), for biomedical applications has been of high interest to researchers for several decades. High chemical photostability, spectral characteristics (high absorption in the red spectral region and fluorescence quantum yield), diverse coordination properties, and architectural flexibility make Pcs one of the most widely used macrocyclic and coordination compounds.

However, the almost complete insolubility of unsubstituted Pcs in water due to the hydrophobicity of the aromatic core and the flat structure of the molecule [1,2,3], as well as weak (usually less than 1% by weight) solubility in most universal organic solvents, such as sulfolane, dimethyl sulfoxide (DMSO), and tetrahydrofuran [3], significantly limits the scope of biomedical applications of Pcs as fluorescent dyes and PSs [4,5].

Among Pc PSs, ZnPc is the most studied. The high interest in this compound is due to the most efficient ^1^O_2_ generation ΦΔ = 0.53 for photocyanine [6]. AlPc is the second-most effective compound for ^1^O_2_ generation ΦΔ = 0.42 for photosens [7]. However, the phenomena of fluorescence and photodynamic activity when entering cells is not observed for ZnPc NPs without substituents making them water-soluble, since the crystalline packaging of ZnPc and AlPc molecules is different, being H-aggregate and J-aggregate, respectively.

This work is devoted to the study of the spectroscopic properties of AlPc aggregates, which can be used as an alternative to molecular solutions of PS for PDT in order to increase accumulation selectivity and tumor/normal contrast.

To increase the selectivity of Pc-based PS accumulation in tumor tissue, many approaches have been developed in recent years [8,9,10], both passive (using lipid and polymer micelles [11,12] and liposomes [13,14,15] as delivery vehicles; conjugation of Pc molecules and gold nanoparticles [16]) and active targeting (functionalization of the surface of carrier nanoparticles with targeting agents—monoclonal antibodies [17]). The implementation of these methods has shown an increase in the selectivity of the accumulation of Pc nanocomplexes in the target tissue of severalfold [12].

It should be noted that for conjugates and self-aggregated Pc complexes, a change in optical spectral properties is observed. The difference between the absorption spectra of isolated and aggregated Pc molecules is related to the dipole–dipole interaction within the framework of the exciton coupling theory [18,19]. The relative orientation of the dipole moments associated with neighboring molecules causes the rearrangement of electron clouds, thus generating new electronic states with different energy levels compared to individual molecules. Pc aggregates often have a red shift of the main absorption peak (Q-band) in comparison with monomers in solution. In addition, a general broadening of the absorption peak is observed, indicating the actual occurrence of multiple interactions among neighboring molecules in the aggregate [20].

In addition, the lifetimes of the triplet and singlet states (phosphorescence and fluorescence, respectively) of conjugates and self-aggregated complexes can change upon interaction with the microenvironment. Antunes et al. [21] observed a change in fluorescence lifetime of ZnPc depending on the geometric arrangement of the molecule relative to the surface of FePc NPs. J.A. Lacey and D. Phillips [22] showed that the fluorescence lifetime of aluminum phthalocyanine disulfide AlPcS_2_ varies depending on the type of bacteria incubated with it. Unlike the pure AlPcS_2_ solution with single-exponential fluorescence kinetics, the AlPcS_2_ incubated with bacteria showed the presence of two components in the decay kinetics: a short lifetime from 0.25 to 1.50 ns, and a long lifetime from 4.79 to 6.15 ns. E. Yaghini et al. [23], who studied the fluorescence lifetime of PEGylated CdSe/ZnS quantum dots conjugated with amphiphilic AlPcS_2_, found that when ingested into the MCF-7 cell culture, there was a change in average lifetime from 3.8 ns to 1.8 ns. As in the study of J.A. Lacey and D. Phillips, the contribution of the short component of the fluorescence lifetime changed the most—from 2.1 ns to 0.8 ns.

The nature of the mechanism of changing Pcs fluorescence lifetime was studied by S. Dhami [24], where it was shown that with a change in the phospholipid concentration in the liposome-forming system, the fluorescence attenuation curve changed from monoexponential (at very low concentrations of phospholipid in water, 1:5000) to two-exponential (with an increase in the concentration of phospholipid) and became monoexponential again when the water concentration in the phospholipid became low. That is, depending on the composition of the medium (lipids/water) surrounding the AlPcS_2_ molecule, the fluorescence lifetime of the molecule also changes.

A. M. Garcia et al. [25] compared DPCC liposomes and BSA as ZnPc-delivery systems into HeLa cells. Different localization of PS in the cell was associated with differences in the fluorescence lifetime: with liposomal delivery, the maximum fluorescence was observed in the cytoplasm and on the membrane, and the fluorescence lifetime of ZnPc was 2.0 ns, while with BSA delivery, the maximum fluorescence was observed in the cytoplasm and nucleus, the lifetime being 3.8 ns, which was considered by the authors to be aggregation of PS after its accumulation in cells.

When Pcs are conjugated with other NPs or self-aggregate, a decrease in the fluorescence quantum yield and ^1^O_2_ generation is usually observed due to due to the high rate of excited state deactivation by nonradiative processes [26,27]. However, the conjugation of Pcs with gold NPs demonstrated an effect of enhanced ^1^O_2_ generation by increasing the quantum yield of the triplet state and energy transfer from the gold NPs to the Pc molecule [28]. The effect of plasmon-enhanced fluorescence and ^1^O_2_ generation of AlPc conjugated with gold NPs has been well studied [29,30,31,32]. A threefold increase in the generation of reactive oxygen species, a sixfold increase in cellular uptake, and more than a tenfold increase in phototoxicity in N-TiO_2_-AlPc NPs compared to molecular AlPc were demonstrated on HeLa and KB cell cultures [33]. The advantages of using nanophotosensitizers based on solid lipid particles loaded with AlPcCl were studied by P.L. Goto et al. [34], where a threefold increase in phototoxicity of N-TiO_2_-AlPc compared to the molecular form of AlPc on melanoma B16-F10 cell cultures was shown.

We have previously discovered the effect of changes in the absorption and fluorescence spectra, the dynamics of interaction with laser radiation (photobleaching), and the fluorescence lifetime of self-aggregated AlPcCl nanoparticles upon changes in colloid pH, capture by immune cells (monocytes [35] and macrophages [36]), interaction with bacteria [37,38], and ingress of these particles into inflamed biological tissue [35,39]. We suggest that registration and characterization of changes in the spectral properties of AlPc NPs accumulated by malignant or tumor-associated cells can make it possible to assess the direction of the tumor process.

The aim of this work was to study the spectroscopic properties of AlPc aggregates in various microenvironments in order to fundamentally assess the possibility of their use as an alternative to molecular solutions of Pc photosensitizers.

## 2. Materials and Methods

### 2.1. Preparation and Characterization of Nanoparticle Colloids

To prepare samples of colloids for research, we used coarse particles (10–20 μm) of AlPcCl (NIOPIK, Russia).

A suspension of AlPc in distilled water (1 mg/mL) was subjected to ultrasonic dispersal for 20 min using a Sonopuls HD 2070 ultrasonic homogenizer (Bandelin, Berlin, Germany) with a KE76 nozzle (20 kHz, amplitude 165 μm). Extraction of small fractions was carried out by centrifugation of the obtained colloids for 10 min at 15,000 rpm.

The average hydrodynamic particle radius and size distribution were determined using a dynamic light-scattering spectrometer (Photocor, Moscow, Russia). The zeta potential was determined using a Zetasizer (Malvern Instruments, Malvern, UK).

The scanning electron microscope NVision 40 (Carl Zeiss, Oberkochen, Germany) was used to determine the shape of the resulting nanoparticles. The crystal packing type of the obtained nanoparticles and AlPc NPs in various biological media (saline solution, blood serum, RPMI cell medium, phosphate-buffered saline (PBS)) was determined using X-ray diffraction on Bruker D8 Advance diffractometer (Bruker, Billerica, MA, USA). X-ray diffraction patterns were analyzed using PCPDS PDF2 database of powder diffraction patterns. X-ray images were processed using Difwin1 (LLC Etalon PTC, Moscow, Russia) and Powder 2.0 (University of Paris XI, Paris, France) software.

The absorption spectra of the obtained colloids in the ultraviolet, visible, and near-infrared (NIR) regions (350–900 nm) were measured using a U-3400 spectrophotometer (Hitachi, Tokyo, Japan) in 1 mm-thick quartz cuvettes.

To measure the fluorescence lifetime, we used a system consisting of a C9300 streak camera and a C10627 streak scope (Hamamatsu Photonics, Hamamatsu, Japan) with picosecond resolution. A 637 nm picosecond semiconductor laser (Hamamatsu Photonics, Hamamatsu, Japan) was used as an excitation source.

Fluorescence spectra were recorded using a LESA-01 spectrum analyzer (BioSpec, Moscow, Russia) with excitation by 405 and 633 nm CW lasers (BioSpec, Moscow, Russia).

In order to correctly determine the concentration of AlPc NP colloids obtained after centrifugation, a calibration curve was constructed. For this purpose, the attenuation spectra of AlPc NP colloids at concentrations of 10, 20, 50, 100, and 250 mg/L were measured. Millimeter-long cuvettes were used for the measurements, which allowed the measurements to be made in the single-scattering approximation.

To interpret the behavior of AlPc NPs in lysosomes, colloids were prepared in distilled water at various pH values. To bring the pH to the required value, a 1% solution of HCl or NaOH was added to the resulting colloids to obtain colloids with pH 2, 3, 4, 10, and 12. The final pH value of the colloid was monitored using an FP20 standard pH meter (Mettler Toledo, Columbus, OH, USA). Absorption spectra, fluorescence spectra, and fluorescence kinetics were measured 60 min after adjusting the pH.

### 2.2. Assessment of Intracellular Accumulation and Localization

The following lines were selected for in vitro studies on cell cultures: HeLa—cervical cancer cells, THP-1—human monocytic cell line derived from an acute monocytic leukemia patient (suspension). In addition, we used a culture of polarized macrophages obtained from isolated human peripheral blood monocytes by adding to the culture medium such factors as interferon-gamma (for polarization in M1—proinflammatory macrophages), interleukin 4 (for polarization in M2—repairing macrophages) and without additional factors (M0—nonactivated macrophages) according to the method proposed by Grachev [40]. These cells play a crucial role in the oncogenesis and pathogenesis of chronic inflammatory diseases, such as rheumatoid arthritis.

Cells in equal amounts were seeded on 35 mm petri dishes, with colloids or solutions added after 1 day, when the cell density was almost monolayered. After incubation for various periods, the cells were washed and centrifuged. The fluorescence was measured from these sedimented cells after addition of DMSO (approximately 20 μL of cell mass and 100 μL DMSO). The dependence of fluorescence intensity of the cell mass dissolved in DMSO was plotted as a function of incubation time.

The intracellular distribution of AlPc NPs was studied using an LSM-710 laser-scanning confocal microscope (Carl Zeiss, Jena, Germany). A 633 nm laser was used to excite the fluorescence of AlPc NPs. Fluorescence spectra were recorded using a 32-channel GaAsP detector. To establish colocalization, we used staining with the fluorescent dye LysoTracker^®^ Green DND-26 (Molecular Probes™, Eugene, OR, USA), which selectively accumulates in lysosomes.

AlPc NP colloids at a concentration of 11 µg/mL were added to the cultural medium for 24 h to prepolarized macrophages growing on coverslips. Fluorescence spectra were recorded from a cell monolayer upon excitation with a 633 nm laser.

A study of the fluorescence kinetics of AlPc NPs located inside cells was carried out on HeLa and M0, M1, and M2 macrophage cell cultures. Analysis of the dynamics of changes in the crystal structure of AlPc NPs inside cells was carried out on HeLa cancer cells after 1 and 24 h of cell incubation with AlPc NPs.

### 2.3. Estimation of Fluorescence Energy Yield and Photodynamic Efficiency

Experimental setups and methods for estimation of fluorescence energy yield and photodynamic efficiency of aluminum phthalocyanine nanoparticles are shown in Appendix A.

Assessment of the photodynamic efficiency of AlPc NPs was carried out by recording the changes in the absorption spectra of hemoglobin during deoxygenation (Appendix A) [41].

When PS is irradiated with laser into the absorption band, energy is transferred from PS to molecular oxygen in the system and its transition to the singlet state occurs, followed by chemical quenching, as a result of which the concentration of molecular oxygen in the system decreases. Hemoglobin is an additional source of oxygen; therefore, to restore the equilibrium concentration, hemoglobin releases stored oxygen into the system, and hemoglobin deoxygenation occurs. This dependence can be approximated by the function:(1)$$y\left(t\right)=A\cdot e^{-kt}+y_{0}$$
where *y*_0_ is the final value of hemoglobin oxygenation in the sample after the completion of irradiation, *t* is the time of laser exposure, and *k* is the photodynamic efficiency coefficient.

An example of the resulting dependence of hemoglobin oxygenation in the sample on the time of laser exposure is shown in Appendix A. The photodynamic efficiency of various PSs will be proportional to the coefficient *k*. Different PSs can be compared in terms of photodynamic efficiency under the same conditions of irradiation and registration of spectra (with the same measurement geometry).

To record the absorption spectra of the studied samples, the LESA-01 spectrometer was used. The measurement scheme is shown in Appendix A. To record the absorption spectra of hemoglobin, a broadband tungsten–halogen light source was used. A 675 nm semiconductor laser (BioSpec, Moscow, Russia) was used as a source for excitation of PS. The power density of laser radiation on the sample surface was 200 mW/cm^2^.

The photodynamic efficiency of AlPc NPs was evaluated on cultures of HeLa cancer cells and human THP-1 monocytes. As a positive control, a comparison was made with the photodynamic efficiency of Photosens (NIOPIK), approved for medical use in Russia (a mixture of water-soluble di-, tri-, and tetrasulfonated AlPc). Each PS was added to the nutrient medium. The incubation time was 24 h for AlPc NPs and 4 h for Photosens.

Two types of samples were prepared:Photosens + cells (HeLa or THP-1) + erythrocytes. The final concentration of Photosens in the medium was 20 mg/L (0.23 μM).NP-AlPc + cells (HeLa or THP-1) + erythrocytes. The final concentration of NPs in the medium was 25 mg/L (the total molar concentration of AlPc in the sample was 0.43 μM).

The energy yield of AlPc NP fluorescence in cell cultures was evaluated in HeLa cells. The energy yield of AlPc NPs was also compared with the energy yield of Photosens in cells. The fluorescence spectra, which were subsequently used to calculate the energy yield, were measured using the method and experimental setup described in [42] (Appendix A). The sample was placed in a modified integrating sphere (Avantes, Apeldoorn, Netherlands), and excitation laser radiation (633 nm) was introduced into the sphere and focused on the sample. The laser radiation and fluorescence scattered inside the sphere were collected into a fiber and measured by the LESA-01. The energy yield was calculated as the ratio of the power emitted in a range of 650–750 nm to the power absorbed at an excitation wavelength of 633 nm. The power absorbed by the test sample was measured as the difference between the power of the scattered laser radiation from the reference unabsorbent sample and the test sample. Cells of the same volume without the PS addition were used as a reference unabsorbent sample.

## 3. Results

### 3.1. Dimensional and Spectroscopic Characteristics of the Obtained AlPc NP Colloids

Figure 1 shows the results of the AlPc NP colloids characterization by size, absorption and fluorescence spectra, and calibration curves for estimating the mass concentration of particles in the colloids after centrifugation.

After ultrasonic dispersal (USD), the average hydrodynamic particle size in the colloids was 400 ± 2 00 nm (Figure 1c). When centrifuged at 15,000 rpm for 15 min, 104 ± 54 nm particles remained in the supernatant (Figure 1c). SEM also confirmed that the average particle size was 100 nm (Figure 1e).

Extinction spectra in the range of 350–900 nm for AlPc NP colloids with concentrations of 10–250 mg/L are presented in Figure 1f (left). A calibration curve for determining the NP concentration obtained by optical density (OD) versus AlPc NP mass concentration from 10 to 250 mg/L obtained after centrifugation is shown in Figure 1f (right). Since the measurements were carried out in the single scattering approximation, the OD for different wavelengths (400, 532, and 755 nm) depended almost linearly on the concentration.

The obtained AlPc NP colloids, in contrast to the water-soluble sulfonated form of AlPc (Photosens), did not fluoresce and had an absorption spectrum characteristic of crystalline AlPc (Figure 1d).

Due to their planar structure (Figure 1a), AlPc molecules can form molecular crystals. SEM images and X-ray pattern of AlPc coarse-particle powder are presented in Figure 2b and Appendix A. AlPc microparticles are molecular crystals of the triclinic syngony of the P1 space group with the lattice parameters shown in Appendix A.

### 3.2. Changes in the Spectral Characteristics of AlPc NPs Depending on the pH

It is known that AlPc NPs begin to fluoresce in a biological environment [37,38], and the fluorescence intensity depends on the type of environment. One of the assumptions was that the appearance of fluorescence is affected by the pH of the surrounding medium. The uptake of NPs mainly occurs by endocytosis, as a result of which the NPs end up in endosomes, which then merge with lysosomes. Lysosomes have an internal pH of 4.5 and are responsible for the breakdown of substances that have entered the cell in both healthy and malignant tissue. The obtained dependence of the absorption and fluorescence spectra (excitation at 633 nm) and characteristics of the fluorescence lifetime of AlPc NPs at pH 2, 3, 4, 7, 10, and 12 are shown in Figure 2.

At pH 2, pH 3, and pH 12, an absorption peak is observed in the region of 650–700 nm, which is absent at pH 7 and is probably associated with the electrostatic interaction of AlPc NP surface molecules with free charge carriers in the colloids or was due to some individual molecules dissolved in the solution. The most intense fluorescence was recorded at pH 2. At pH 12, the fluorescence intensity was four times less, and at neutral pH 7 it was almost completely absent (Figure 2b). At pH 4 and pH 10, fluorescence was also observed, but its intensity was significantly reduced. It should be noted that, depending on pH, the absorption and fluorescence maxima shifted by 9 and 12 nm, respectively. At acidic pH, a significant broadening of the fluorescent peak was observed, which may be due to conformational changes in individual AlPc molecules, the presence of aggregates, or energy transfer between NPs and the AlPc molecule.

An analysis of the fluorescence kinetics of AlPc NPs (Figure 2d) showed the presence of two lifetime components: τ_1_ = 0.43 ns and τ_2_ = 5.64 ns at pH 4 and τ_1_ = 0.68 ns and τ_2_ = 4.19 ns at pH 10. At more acidic and alkaline pH, only one lifetime component, τ_2_ ≈ 5 ns, was observed, which corresponds to the lifetime of individual AlPcCl molecules. The appearance of shorter fluorescence lifetimes can be explained by the presence of dimers in the colloids or by the presence of an electrostatic bond between the quasi-free AlPc molecules and NPs.

### 3.3. AlPc NP Accumulation, Localization, and Fluorescence in Cells

Figure 3 shows the data on the uptake of AlPc NPs by HeLa cells compared to Photosens by fluorescence intensity with excitation at 405 nm. The fluorescence spectra of NPs inside HeLa cells correspond to the fluorescence spectrum of individual AlPc molecules (Figure 3a). However, the fluorescence intensity is small compared to the intensity of Photosens (Figure 3b). When DMSO solvent, which dissolves both cells and NPs in them, was added to the cell sediment, it can be seen that the amount of accumulated Pc during the incubation with NPs was even greater than the accumulation of Photosens (Figure 3c). Thus, cells have a “supply” of nanoparticles that do not exhibit photoactive properties, but they can appear when conditions change during the life of the cell.

It has been demonstrated (Appendix A) that individual diffraction peaks persist inside the cells, especially strongly at 7°, which corresponds to the crystal plane with indices (001). Thus, inside cells, AlPc NPs retain their crystalline structure (short-range order of symmetry) without passing into an amorphous state, and also without completely dissolving into individual molecules. This fact is confirmed by the absorption spectra measured from the cell mass with accumulated AlPc NPs inside, corresponding to the absorption spectrum of the AlPc NP water colloids (Appendix A).

The results of the study of intracellular AlPc NPs localization in HeLa cell culture after 24 h of incubation are shown in Figure 4. Cell lysosomes were preliminarily stained with Lysotracker. On the first scan with a 633 nm laser to obtain a microscopic image, the fluorescence signals of the lysosomal dye and AlPc NPs are colocalized. Long-term exposure of cells to 633 nm laser radiation destroyed lysosomes that had accumulated AlPc NPs, photobleached the lysosomal dye due to the photodynamic activity of AlPc NPs, flared up NP fluorescence at a wavelength of 682 nm, and spread photoactive AlPc NPs throughout the volume of the cell cytoplasm. We suggest that the occurrence of AlPc NP fluorescence is associated with an acidic environment and a large amount of soluble hydrolytic enzymes in the lysosome interior. With further laser exposure, a decrease in fluorescence intensity is observed, which may be due to either photobleaching or rapid aggregation of the molecules. No such effect is observed for Photosens.

Similar behavior of AlPc NPs was observed when interacting with macrophages of various phenotypes: M0, M1, and M2 (Figure 5). After incubation of colloids with cells, AlPc NPs accumulated inside the cells begin to fluoresce. Confocal microscopy of the intracellular distribution demonstrated accumulation in lysosomes. Also, a noticeable photobleaching of AlPc NPs in cells was observed after laser exposure, which was absent for the sulfonated form of AlPc.

The study of the AlPc NP fluorescence decay kinetics when interacting with macrophages reveals a change in the fluorescence lifetime (Figure 6). The difference between the fluorescence decay kinetics shows that the lifetime of AlPc NP fluorescence depends on the polarization of macrophages, and based on our data, we can assume that AlPc NPs have at least two possible states, each of which differs in the nature of the interaction of molecules on the surface of the nanoparticle with the environment. We associate the differences in the fluorescence properties of AlPc NPs when they enter macrophages of different phenotypes with the difference in the biochemical composition of macrophage lysosomes [43]. The results of the AlPc NP fluorescence lifetimes in macrophages are shown in Figure 6 on the right.

### 3.4. Photodynamic Efficiency and Fluorescence Energy Yield of AlPc NPs upon Interaction with Cells

Figure 7 shows the experimental time dependence of the hemoglobin deoxygenation obtained on HeLa and THP-1 cell samples incubated with Photosens and AlPc NPs.

Pcs and in particular AlPc have been shown to be efficient type II sensitizers in simple chemical systems. However, in more complex systems, for example in cells, the photodynamic reaction involving Pcs also proceeds according to type I [44]. The measurement method used does not allow one to determine the quantum yield of ^1^O_2_ generation; however, it makes it possible to compare the oxygen-dependent photodynamic efficiency of PSs.

From the obtained data, it can be concluded that in the case of AlPc NPs, the photodynamic efficiency coefficient in HeLa cell cultures is 2.5 times lower than Photosens. The photodynamic efficiency of AlPc NPs in THP-1 cell cultures at the initial stage of irradiation exceeds the efficiency of Photosens; however, with further irradiation, a decrease in the rate of hemoglobin deoxygenation is observed. This is probably due to the ability of THP-1 monocytes to act on captured nanoparticles and transfer them to a phototoxic state. Irradiation causes deactivation of the active AlPc NP molecules and the destruction of lysosomes due to the photodynamic effect, which leads to cell death.

The scattered laser radiation and fluorescence spectra of HeLa cells with AlPc photosensitizer in molecular form and nanoform are shown in Appendix A. The calculated energy yield of fluorescence for Photosens and AlPc NPs in cells was 0.13 and 0.01, respectively. From literature data, the quantum yield of fluorescence for AlPcCl in ethanol is 0.52 [45] and for Photosense in water 0.40 [46]. Such a low fluorescence energy yield for AlPc NPs may be due to the fact that not all molecules constituting NPs, but only surface ones, are capable of fluorescence. It is also worth noting that the PS fluorescence energy yield inside cells can decrease due to an increase in the rate of nonradiative relaxation of the PS excited state due to the proximity of various quenchers.

### 3.5. AlPc NP Fluorescence and Phototoxicity Mechanism

The research results show that AlPc NPs have interesting spectroscopic properties within cells that may be applicable for phototheranostics. At the same time, their therapeutic selectivity for the destruction of target biological structures is higher than that of a clinically approved photosensitizer based on AlPc (Photosens). This is achieved due to the fact that, under photodynamic action, the transition of AlPc NPs to a nonphototoxic state occurs during the destruction of cells and cellular structures where they are accumulated. The high selectivity of accumulation in tumor, tumor-associated, and immunocompetent (macrophage) cells compared to healthy cells will be provided by the enhanced penetration and retention effect [47,48,49] and naturally high endocytosis activity [50].

An urgent problem that will need to be solved in the near future is the modification of the hydrophobic surface of AlPc NPs to enable intravenous administration. However, this should not be an obstacle for their local administration for phototheranostics of inflammatory diseases, for example, in osteoarthritis [51] and assessment of tissue engraftment during transplantation [35,39]. The sensitivity of the fluorescence lifetime of AlPc NPs to the environment can make it possible to recognize macrophages of different phenotypes, as well as to photodynamically influence them.

However, the retention of crystallinity, the absence of an absorption peak of the molecular form of AlPc in cells, the appearance of several fluorescence lifetimes, and the high rate of photobleaching under laser irradiation raise the suspicion of the fluorescence and phototoxicity occurrence in cells only of the NP dissolution into individual phototoxic molecules. One of the possible mechanisms to explain the above properties may be the interaction of AlPc NPs with surrounding cellular biomolecules (lysosomal hydrolytic enzymes and membrane proteins [52,53]), in which the structural rearrangement of the surface molecules of AlPc NPs occurs (Figure 8). That is, surface molecules with greater mobility than the bulk part can, when interacting with the external environment, transition from being a part of the NP crystal structure to a state similar to that of a free molecule (quasi-free). Being held by one or more bonds with other surface molecules (members of the crystal lattice), they exhibit the spectroscopic properties of free molecules, while they can effectively exchange energy both with the crystal and other neighboring molecules that have passed into a quasi-free state. In this position, surface molecules are able to fluoresce and transfer energy to molecular oxygen, transforming it into a chemically active singlet state. As a result, the chemical bond between AlPc and biomolecule is destroyed and the NPs pass into a nonfluorescent and nonphototoxic state. Unfortunately, we did not find a way to unambiguously confirm this hypothesis; however, all indirect signs point to the possibility of such an interaction.

The experimental data obtained from XRD analysis, absorption spectra, and fluorescence kinetics of AlPc NPs in cells prove the proposed mechanism for the occurrence of AlPc NP fluorescence and phototoxicity.

## 4. Conclusions

Increasing phototheranostic specificity and selectivity remains an urgent task. The use of AlPc NPs as photosensitizers can significantly improve the selectivity of photodynamic therapy.

In this work, colloids of crystalline AlPc NPs with dimensions of 104 ± 54 nm were obtained and a method for determining the mass concentration of particles in a colloid was developed. It was shown that when the pH changes, the absorption spectrum of NPs changes, and fluorescence appears, which corresponds to the AlPc monomer spectrum, but differs in lifetime. At pH 4 and pH 10, the fluorescence kinetics of AlPc NPs showed the presence of two lifetime components: τ_1_ = 0.43 ns and τ_2_ = 5.64 ns and τ_1_ = 0.68 ns and τ_2_ = 4.19 ns, respectively.

The accumulation of AlPc NPs in the lysosomes of HeLa cells and macrophages and the appearance of fluorescence with different kinetics while maintaining crystallinity were shown. Differences in the dynamics of changes in the fluorescence intensity of AlPc NPs in lysosomes of HeLa cells under laser irradiation (flare-up and photobleaching) compared to Photosens were demonstrated.

The fluorescence quantum yield of AlPc NPs in HeLa cells was 1% compared to 13% for Photosens. It was shown that the photodynamic efficiency *k* in HeLa cells is 1.3 times higher in Photosens, while NP-AlPc is 1.8 times more effective for THP-1.

The retention of crystallinity, the absence of an absorption peak of the molecular form of AlPc in cells, the appearance of several fluorescence lifetimes, and the high rate of photobleaching under laser irradiation raise the suspicion of the fluorescence and phototoxicity occurrence in cells only of the NP dissolution into individual phototoxic molecules. Various structural, spectral, and fluorescent characteristics of NP-AlPc and Photosens when interacting with cells forced us to propose our own mechanism for the interaction of NP AlPc with cells, but unfortunately, we did not find a way to univocally confirm this hypothesis.

Thus, the fundamental possibility of using crystalline AlPc NPs for the purposes of phototheranostics has been demonstrated. The use of AlPc NPs as photosensitizers can significantly increase the specificity and selectivity of phototheranostic methods due to the possibility of recognizing the predominant macrophage phenotype in tissue, as well as the absence of AlPc NP phototoxicity before being distributed through lysosomes. Further research will focus on a more detailed study of theranostic properties of AlPc NPs, such as the efficiency of singlet-oxygen generation in cells, and EC_50_ that shows efficiency of photosensitizer on cells and other living systems, again to enable comparison with clinically used or other potential photosensitizers.

## Figures and Tables

**Figure 1 pharmaceutics-14-02122-f001:**
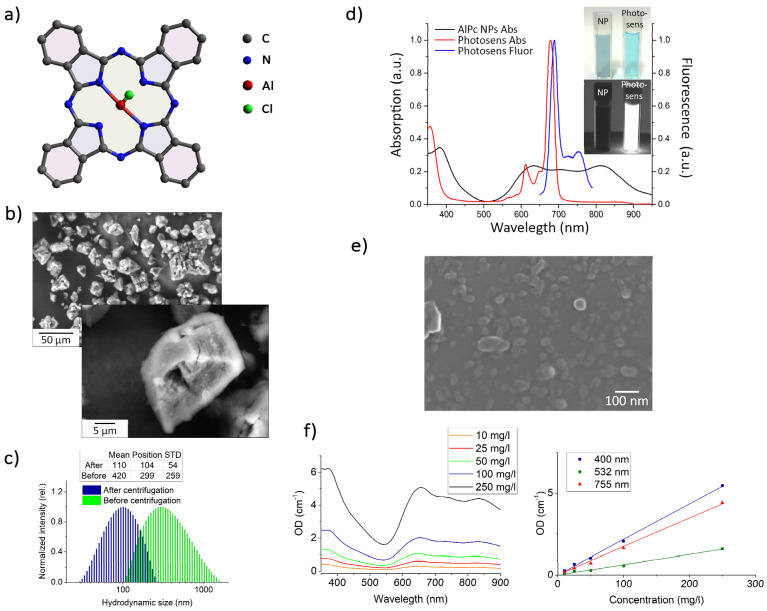
(**a**) Aluminum phthalocyanine chloride (AlPcCl) used in the study; (**b**) SEM images of AlPc microparticles; (**c**) size distribution of the obtained particles in the colloid before and after centrifugation; (**d**) absorption spectra of Photosens and of crystalline AlPc NP colloids and the fluorescence spectrum of Photosens, as well as their visible and fluorescent images at 400 nm excitation (inset) (**e**) SEM images of obtained AlPc NPs; (**f**) absorption spectra for various concentrations of AlPc NP colloids: 10–250 mg/L (left), calibration curves for determining the concentration of AlPc NP colloids (right) by OD.

**Figure 2 pharmaceutics-14-02122-f002:**
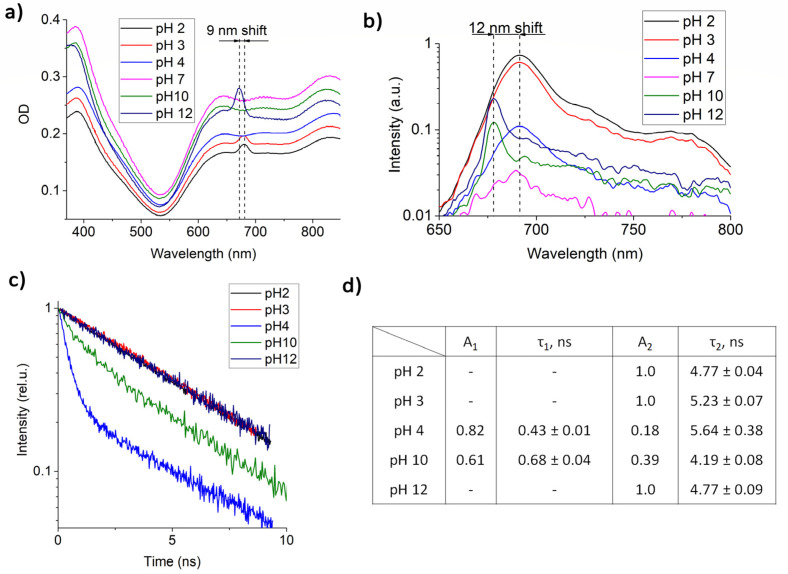
(**a**) AlPc NP absorption spectra at different pH; (**b**) AlPc NP fluorescence spectra (excitation at 633 nm) at different pH (due to the large difference in fluorescence intensity, the spectra are shown on a logarithmic scale); (**c**) AlPc NP fluorescence kinetics at different pH (**d**) calculated fluorescence lifetimes and amplitudes.

**Figure 3 pharmaceutics-14-02122-f003:**
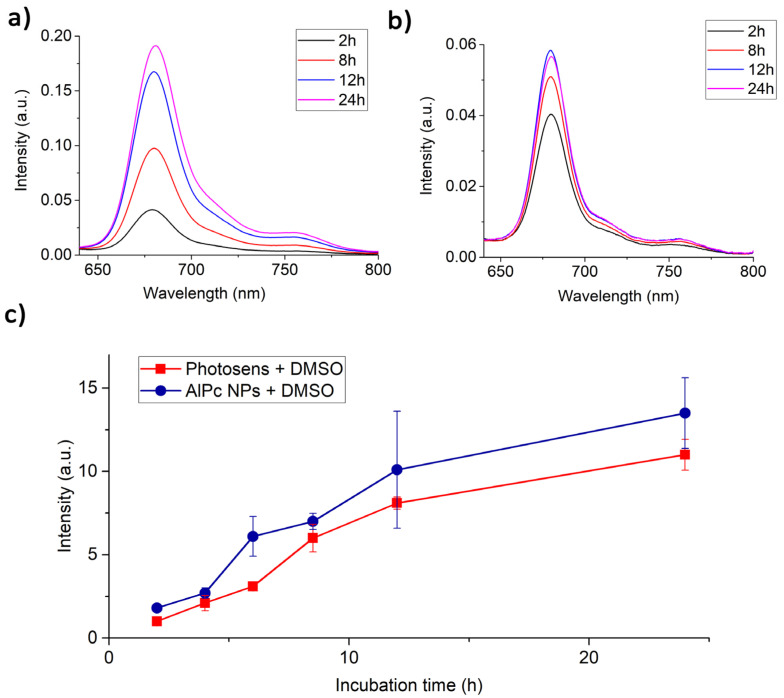
(**a**) Fluorescence spectra of Photosens in HeLa cells depending on the duration of incubation; (**b**) fluorescence spectra of AlPc NPs in HeLa cells depending on the duration of incubation; (**c**) fluorescence intensity of Photosens and AlPc NPs with addition of DMSO in HeLa cells.

**Figure 4 pharmaceutics-14-02122-f004:**
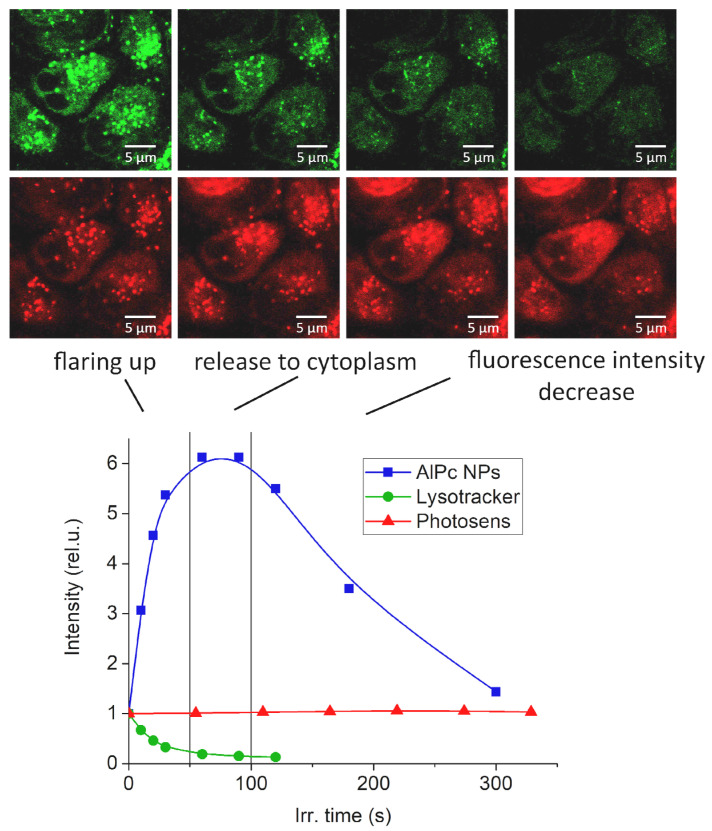
Fluorescence images of HeLa cells (top row) after 24 h of incubation with AlPc NPs. Top row: green—Lysotracker fluorescence, red—AlPc NP fluorescence, from left to right—destruction of lysosomes by irradiation with a 633 nm laser. Bottom: fluorescence-intensity dynamics during laser irradiation.

**Figure 5 pharmaceutics-14-02122-f005:**
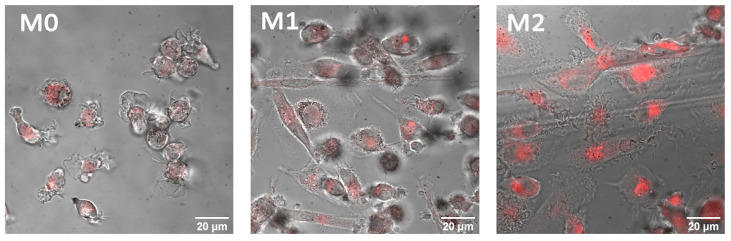
Fluorescence images of M0, M1 and M2 macrophages after 24 h of incubation with AlPc NP colloids. AlPc NP fluorescence is red.

**Figure 6 pharmaceutics-14-02122-f006:**
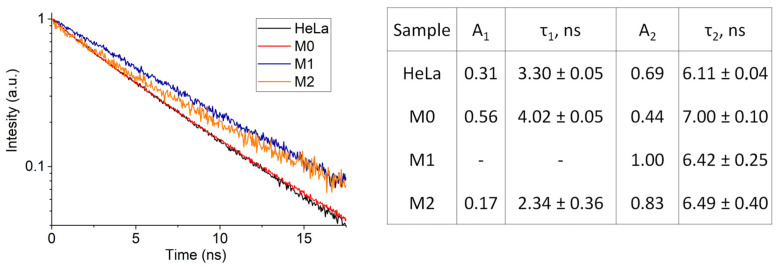
Fluorescence kinetics of AlPc NPs upon interaction with M0, M1 and M2 macrophages and calculated fluorescence lifetimes and amplitudes.

**Figure 7 pharmaceutics-14-02122-f007:**
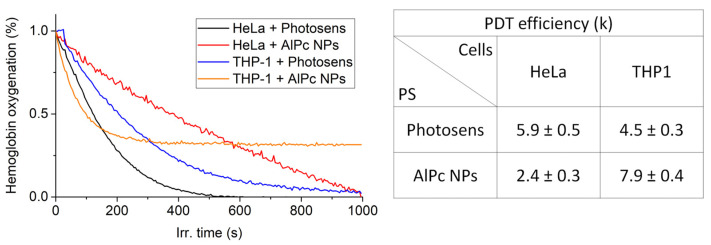
Dependence of hemoglobin oxygenation in the sample on the duration of laser irradiation at a wavelength of 675 nm with a power density of 200 mW/cm^2^ for HeLa and THP-1 cells incubated with AlPc NPs and Photosens, as well as the calculated values of photodynamic efficiency coefficient (k).

**Figure 8 pharmaceutics-14-02122-f008:**
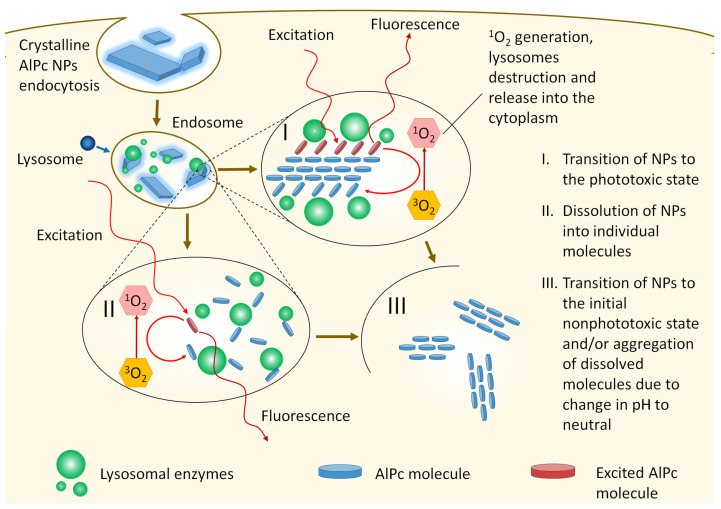
Model of the fluorescence and phototoxicity mechanism of AlPc NPs captured by cells.

## Data Availability

Not applicable.

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
