# Peer review of "Theranostic Properties of Crystalline Aluminum Phthalocyanine Nanoparticles as a Photosensitizer"

_pharmaceutics, 2022, doi:10.3390/pharmaceutics14102122_

Round 1

Reviewer 1 Report (Previous Reviewer 1)

Review

The manuscript submits an interesting study. The information contained could be of concern especially to scientists I the field of nanomaterials. A certain shortcoming is the fact that authors used a somewhat older, in some parameters currently overcome molecules. In addition, the data presented do not clearly confirm the declared hypothesis and conclusion stated by authors that the AlPc NPs constructed and tested have “a high potential as phototheranostics” (lines 464 and 465).  Anyway the approach used can be useful also for other structures.

Notes and questions to the text:

Lines 276 and 280: t and k, respectively, should be in italic.

Line 299: Why the incubation times are different? Are the related results of the AlPcs NPs and Photosens comparable?

Line 305: The value 0.43 uM is a concentration of complete NPc or AlPc bound in NPs? It should be clarified.

Fig 3 and corresponding text: Is the photophysical behaviour (absorption peaks at pH 2 and 12 and the highest fluorescence at the same conditions) reversible after changing the pH again? It should be noted as it can indicate (in)stability of the NPs construct.

Figure 4: Why 12 h only is the max time in case of b), unlike of a) and c), where the max time is 24h? Couldn´t it affect the results?

Author Response

The authors express their deep gratitude to reviewer for the work done!

The manuscript submits an interesting study. The information contained could be of concern especially to scientists I the field of nanomaterials. A certain shortcoming is the fact that authors used a somewhat older, in some parameters currently overcome molecules. In addition, the data presented do not clearly confirm the declared hypothesis and conclusion stated by authors that the AlPc NPs constructed and tested have “a high potential as phototheranostics” (lines 464 and 465).  Anyway the approach used can be useful also for other structures.

We have corrected the wording to match the results presented.

Notes and questions to the text:

Lines 276 and 280: t and k, respectively, should be in italic.

The font style has been fixed

Line 299: Why the incubation times are different? Are the related results of the AlPcs NPs and Photosens comparable?

The incubation time was the same. Figure 4 (now 3) has been corrected and the measurement times have been unified.

Line 305: The value 0.43 uM is a concentration of complete NPc or AlPc bound in NPs? It should be clarified.

This is the total concentration of AlPc NPs in the sample. We have added explanatory text to the article.

Fig 3 and corresponding text: Is the photophysical behaviour (absorption peaks at pH 2 and 12 and the highest fluorescence at the same conditions) reversible after changing the pH again? It should be noted as it can indicate (in)stability of the NPs construct.

This effect is reversible, since individual AlPc molecules, if separated from NPs, quickly aggregate and form NPs.

Figure 4: Why 12 h only is the max time in case of b), unlike of a) and c), where the max time is 24h? Couldn´t it affect the results?

Figure 4 (now 3) has been corrected and the measurement times have been unified.

Some parts of the article have been changed in accordance with the comments of other reviewers. Their comments and questions, as well as our answers to them, you can see in the attached file.

Reviewer 2 Report (New Reviewer)

The manuscript is entitled "Theranostic properties of crystalline aluminum phthalocyanine nanoparticles as a photosensitizer: as a new mechanism for the occurrence of fluorescence and phototoxicity of AlPc NPs in interaction with cells has been proposed by the authors. This study also highlighted the importance of AlPc NPs as photosensitizers, which can significantly increase the specificity and selectivity of phototheranostic methods due to the possibility of recognizing the predominant macrophage phenotype in the tissue as well as the absence of AlPc NPs phototoxicity. Overall, this work raises a number of interesting issues and is well organized and produced high-quality results. So, I recommend it in order to be considered for publication after minor revision.

  1. The brightness of all figures was not bold or clear in image resolution. The authors need to re-check the brightness of the figures in order to ensure easy eye catch up.
  2. In Abstract, line 30, the authors wrote "conclusion," which is not the format of articles. So, authors need to take this part into conclusion rather than leave it in the abstract.
  3. The authors forgot to state the mechanisms, making it impossible to follow up on their findings. The authors should incorporate the flow mechanism to attract the reader’s attention.
  4. The excitation wavelength peak and corresponding emission peak must be provided by the authors.
  5. The authors did not report the quantum yield and failed to explain how the quantum yield related to the standard/reference to the claim. If the authors obtain any quantum yield, they must report it.

Author Response

The authors express their deep gratitude to reviewer for the work done!

The manuscript is entitled "Theranostic properties of crystalline aluminum phthalocyanine nanoparticles as a photosensitizer: as a new mechanism for the occurrence of fluorescence and phototoxicity of AlPc NPs in interaction with cells has been proposed by the authors. This study also highlighted the importance of AlPc NPs as photosensitizers, which can significantly increase the specificity and selectivity of phototheranostic methods due to the possibility of recognizing the predominant macrophage phenotype in the tissue as well as the absence of AlPc NPs phototoxicity. Overall, this work raises a number of interesting issues and is well organized and produced high-quality results. So, I recommend it in order to be considered for publication after minor revision.

  1. The brightness of all figures was not bold or clear in image resolution. The authors need to re-check the brightness of the figures in order to ensure easy eye catch up.

The brightness of the drawings has been increased. When creating a PDF file, the quality of the drawings may be significantly degraded. We have tried to keep the best quality in it. Pictures in high resolution were sent to the editorial office separately.

  1. In Abstract, line 30, the authors wrote "conclusion," which is not the format of articles. So, authors need to take this part into conclusion rather than leave it in the abstract.

The "Conclusion" part of the Abstract has been moved to the general "Conclusion". The paragraph "Conclusion" was updated in accordance with the comments of the reviewers.

  1. The authors forgot to state the mechanisms, making it impossible to follow up on their findings. The authors should incorporate the flow mechanism to attract the reader’s attention.

The discussion of the mechanism was moved to a separate subsection “3.5. AlPc NPs fluorescence and phototoxicity mechanism discussion”. The conclusions were supplemented and presented in accordance with the obtained experimental results.

  1. The excitation wavelength peak and corresponding emission peak must be provided by the authors.

The excitation wavelength for obtaining fluorescence spectra was indicated for each case.

  1. The authors did not report the quantum yield and failed to explain how the quantum yield related to the standard/reference to the claim. If the authors obtain any quantum yield, they must report it.

A link to a method for determining the fluorescence quantum yield has been added. Reference values for AlPc fluorescence quantum yields have been added to subsection 3.4.

Some parts of the article have been changed in accordance with the comments of other reviewers. Their comments and questions, as well as our answers to them, you can see in the attached file.

Reviewer 3 Report (New Reviewer)

The results of TEM not found in figure 2.

Author Response

The authors express their deep gratitude to reviewer for the work done!

The results of TEM not found in figure 2.

Thanks for the note, we have corrected the error in the text. The figure was a SEM image

Some parts of the article have been changed in accordance with the comments of other reviewers. Their comments and questions, as well as our answers to them, you can see in the attached file.

Reviewer 4 Report (New Reviewer)

I carefully read the manuscript by Makarov et al. entitled "Theranostic properties of crystalline aluminum phthalocyanine nanoparticles as a photosensitizer", in which the work aimed to formulate an encapsulated phthalocyanine for the creation of a cancer diagnosis method, as well as for its treatment.

In general, the work, although well written and with very satisfactory English, in my perception, the manuscript is disorganized. Despite this, it seems to me to have some of the "ingredients" necessary for publication.

Major comments:

1) In the abstract, the authors state that "One of the promising types of nanophotosensitizers are crystalline aluminum phthalocyanine nanoparticles (AlPc NP)." (lines 14 and 15). I think it's a dangerous phrase to say since, in a way, it can belittle other classes of types of photosensitizers. Authors should simplify the sentence after this one, indicating why these AlPc NPs are unique compared to other PS NPs.

2) Regarding the introduction:

2a) Too long (two full pages of text), making it difficult to follow the evolution of what the authors intend to convey to the reader. Several paragraphs could be simplified or even removed; some information seems vague to me and should be further explored. As my suggestion, information regarding the clinical application should be removed since only in vitro studies are discussed here. Another subject that aroused my curiosity was that the authors stated that ZnPc and AlPc are efficient in the production of singlet oxygen but do not present quantum yields of production of this ROS.

2b) Authors present alternatives to improve the selectivity of the dye family, such as encapsulation in gold nanoparticles or liposomes. Still, they do not adequately reference each of the other options, leaving a set of bibliographic references only at the end of the sentences.

2c) "When Pcs are conjugated with other NPs or self-aggregate, a decrease in the fluorescence quantum yield and 1O2 generation is usually observed due to an enhanced intersystem crossing effect" (lines 122-124). Are the authors sure about this? I think the intersystem crossing effect is significantly reduced. Hence there is no photosensitizer in the triplet state. Introduce bibliographic references.

2d) Just as a suggestion: the last paragraph of the introduction should link the problem to be solved by the authors and the work presented below.

2e) Why do you not present the molecular structure of the photosensitizer in the manuscript, namely in the introduction? I know it's in the supplementary material, but it seems like something too important to be SM.

3) The experimental section is quite descriptive but, at the same time, confusing. Couldn't it be simplified? Examples of spectra are not supposed to be presented in the experimental section. Please try to adhere only to the methodologies you used to obtain the presented results. Equations must be numbered and not shown in figures but as text. Figure 1, in my opinion, should be moved to supplementary material.

4) The results section has no subsection. Regardless of the techniques used, the results are presented without division. Please create subsections. As the manuscript is, the reader cannot find the results he intends to consult described.

5) Section on the conclusions is very little explored. After many techniques were used and many results presented, the authors gave an incomplete conclusion. This has to be significantly improved before publication and includes a key sentence regarding the deduction that was drawn from each methodology used, as well as a final sentence concluding whether the objective was achieved and, if so or negative, pointing out future directions of how the authors intend to move forward with the work.

Furthermore, the conclusions must demonstrate how the studied AlPc NP can serve as a theranostic method.

Minor comments:

6) The expressions "et al.", "in vitro", “versus”, among others, should be placed in italics throughout the entire manuscript (for example, lines 98, 111, 132, 188, 278);

7) line 119 – singlet oxygen had already been mentioned before, but its molecular formula only appears here;

8) line 117 – what are nanoPcs? PC NPs?

9) Several figures are not referenced in the text. Furthermore, the first figure referenced in the text is Figure 2, which makes no sense. I ask you to reorder. The figures must also be found immediately after the first textual reference. The figure legends directly refer to what the different panels are. A sentence relating to all panels, in general, should be included immediately before the reference to what each of the panels presents.

5) Aesthetically, Figure 2 is not very appealing. In Figure 2, the various panels are divided into squares, but in Figure 3, they are not. Uniform. Figures 1c and 1e need to be enlarged to understand, while Figure 1d is too large.

Having significantly improved the points that I have highlighted, I suggest that this work be published in the Pharmaceutics journal.

Author Response

The authors express their deep gratitude to reviewer for the work done!

I carefully read the manuscript by Makarov et al. entitled "Theranostic properties of crystalline aluminum phthalocyanine nanoparticles as a photosensitizer", in which the work aimed to formulate an encapsulated phthalocyanine for the creation of a cancer diagnosis method, as well as for its treatment.

In general, the work, although well written and with very satisfactory English, in my perception, the manuscript is disorganized. Despite this, it seems to me to have some of the "ingredients" necessary for publication.

Major comments:

1) In the abstract, the authors state that "One of the promising types of nanophotosensitizers are crystalline aluminum phthalocyanine nanoparticles (AlPc NP)." (lines 14 and 15). I think it's a dangerous phrase to say since, in a way, it can belittle other classes of types of photosensitizers. Authors should simplify the sentence after this one, indicating why these AlPc NPs are unique compared to other PS NPs.

We have corrected the wording in order not to belittle other classes of types of photosensitizers

2) Regarding the introduction:

2a) Too long (two full pages of text), making it difficult to follow the evolution of what the authors intend to convey to the reader. Several paragraphs could be simplified or even removed; some information seems vague to me and should be further explored. As my suggestion, information regarding the clinical application should be removed since only in vitro studies are discussed here. Another subject that aroused my curiosity was that the authors stated that ZnPc and AlPc are efficient in the production of singlet oxygen but do not present quantum yields of production of this ROS.

We have shortened the introduction by deleting several paragraphs, including those about clinical use. Singlet oxygen quantum yields for ZnPc and AlPc have been added.

2b) Authors present alternatives to improve the selectivity of the dye family, such as encapsulation in gold nanoparticles or liposomes. Still, they do not adequately reference each of the other options, leaving a set of bibliographic references only at the end of the sentences.

Literature references have been clarified and provided for each method.

2c) "When Pcs are conjugated with other NPs or self-aggregate, a decrease in the fluorescence quantum yield and 1O2 generation is usually observed due to an enhanced intersystem crossing effect" (lines 122-124). Are the authors sure about this? I think the intersystem crossing effect is significantly reduced. Hence there is no photosensitizer in the triplet state. Introduce bibliographic references.

The error has been corrected and a bibliographic reference has been provided. Indeed, during aggregation, the reason for the decrease in the quantum yields of fluorescence and the generation of singlet oxygen is an increase in the rate of nonradiative relaxation of excited states.

2d) Just as a suggestion: the last paragraph of the introduction should link the problem to be solved by the authors and the work presented below.

The last paragraph of the introduction, specifying the aim of the work and linking it to the results presented, was added

2e) Why do you not present the molecular structure of the photosensitizer in the manuscript, namely in the introduction? I know it's in the supplementary material, but it seems like something too important to be SM.

We have added the AlPc molecular structure in Figure 1a.

3) The experimental section is quite descriptive but, at the same time, confusing. Couldn't it be simplified? Examples of spectra are not supposed to be presented in the experimental section. Please try to adhere only to the methodologies you used to obtain the presented results. Equations must be numbered and not shown in figures but as text. Figure 1, in my opinion, should be moved to supplementary material.

Examples of absorption spectra are given simultaneously with the measured absorption spectra of the obtained AlPc NPs. We believe that they are necessary for a clear demonstration of the differences in the absorption of the molecular and nanoforms of AlPc.

The equations were numbered, and Figure 1 has been moved to Supplementary Information

4) The results section has no subsection. Regardless of the techniques used, the results are presented without division. Please create subsections. As the manuscript is, the reader cannot find the results he intends to consult described.

Subsections have been added

5) Section on the conclusions is very little explored. After many techniques were used and many results presented, the authors gave an incomplete conclusion. This has to be significantly improved before publication and includes a key sentence regarding the deduction that was drawn from each methodology used, as well as a final sentence concluding whether the objective was achieved and, if so or negative, pointing out future directions of how the authors intend to move forward with the work.

Furthermore, the conclusions must demonstrate how the studied AlPc NP can serve as a theranostic method.

We took into account your comments and rewrote the Conclusion

Minor comments:

6) The expressions "et al.", "in vitro", “versus”, among others, should be placed in italics throughout the entire manuscript (for example, lines 98, 111, 132, 188, 278);

7) line 119 – singlet oxygen had already been mentioned before, but its molecular formula only appears here;

8) line 117 – what are nanoPcs? PC NPs?

9) Several figures are not referenced in the text. Furthermore, the first figure referenced in the text is Figure 2, which makes no sense. I ask you to reorder. The figures must also be found immediately after the first textual reference. The figure legends directly refer to what the different panels are. A sentence relating to all panels, in general, should be included immediately before the reference to what each of the panels presents.

5) Aesthetically, Figure 2 is not very appealing. In Figure 2, the various panels are divided into squares, but in Figure 3, they are not. Uniform. Figures 1c and 1e need to be enlarged to understand, while Figure 1d is too large.

Minor comments were taken into account and corrected

Having significantly improved the points that I have highlighted, I suggest that this work be published in the Pharmaceutics journal.

Thank you for the work done and for recommending the publication of our study in the journal

Some parts of the article have been changed in accordance with the comments of other reviewers. Their comments and questions, as well as our answers to them, you can see in the attached file.

Round 2

Reviewer 4 Report (New Reviewer)

The authors committed to the changes I suggested to improve the manuscript significantly.

The introduction is more concise, the results chapter is properly sectioned and improved, and the conclusions are greatly enhanced.

Not all of my comments were addressed (namely about the materials and methods, which, in my opinion, remain a little confusing...). However, in general, I can consider that the manuscript is improved and with the desired quality at its publication in the Pharmaceutics journal.

Therefore, for my part, I strongly recommend the publication of the work.

This manuscript is a resubmission of an earlier submission. The following is a list of the peer review reports and author responses from that submission.

Round 1

Reviewer 1 Report

The study concerns in interesting topic and approach, even if working with older, in some parameters surpassed molecules. The results can be especially interesting for scientist working in area of nanoparticles. However, the graphical quality of the manuscript should be improved, and some points explained or added to be suitable for publication.

The Introduction is inadequately long, it should be significantly shortened to explain only the just relevant area. About 1/3 of the current extent would be quite enough.

If used, the text of line 43 should be more precise: ….almost complete insolubility of unsubstituted Pc in water…., otherwise it is not true and in contradiction to the following paragraph.

If used, in the text of lines 152, 155, 162 and 166 the oxygen forms and titanium dioxide should be written with corrects indices.

Generally – the figures in manuscript (unlike of SI) are of very bad quality, hardly readable, sometimes impossible to evaluate. They should be improved.

Concerning the potential of photodynamic activity: it is standardly expressed in ΦΔ values, which enables comparison with clinically used or other published potential photosensitizers. It would be useful to include this value into the manuscript.

EC50 or similar one is the preferred value to show efficiency of photosensitizer on cells or other living systems, again to enable comparison with clinically used or other potential photosensitizers. Is such value available for the tested material?

 Figure 4 a) and b) do not well correlate with the text above, it should be described more precisely.

The naming and descriptions at figures in the SI are a bit confusing. The items should be clearly organized.

Reviewer 2 Report

In the manuscript titled “Unique theranostic properties of aluminum phthalocyanine nanoparticles as a photosensitizer”, the authors seek to determine the conditions under which aluminum phthalocyanine NPs upon contact with malignant/immune cells become phototoxic and compare their photodynamic action and specificity with the corresponding molecular formulation approved for medical use. Although showing some promising PDT effects, these NPs have the problem of “switching off” at neutral pH.

The manuscript is, in general, well written and well supported by appropriate literature. The data provided by a set of well-planned experiments seems sound, and well interpreted and provide meaningful information for those working with nanomaterials for photo-theranostics.

I find the work of interest for the Pharmaceutics’ readers and worth publishing after some modifications:

1 – In my opinion, the Introduction section should be more focused and therefore shortened.

2 – On page 3, lines 100-101, it is written “Phthalocyanine aggregates often have a redshift of the main absorption peak (Q-band) in comparison with the solution.”; If I understand it correctly, it should be “…in comparison with the monomers in solution.”

3 – The quality of most figures is very low and must be greatly improved, especially those containing several panels with microscopic images (Figures 1, 2, 3, and 6). In reverse, figure 4 and figure 5 (bottom row) can be reduced.

4 – On page 7, the Results section should be revised: presenting the data in the same order that figure 2 is organized is advisable for easier reading; a figure 2g is mentioned which does not exist; the paragraph in lines 316-319 is very confusing; the size indicated in the label of figure2b concerning the sizes of AlPc microparticles must be wrong (50 microns instead of 50 nm). Figure 2d representing the TEM image of AlPc NPs does not let us conclude on the prevalence of ca. 100 nm NPs in the sample?

5- Could the absorption bands presented in figure 3a (at pH12 and 2) be due to some individual molecules dissolved in the solution which would be responsible for the fluorescence and similar lifetimes obtained at those pH’s?

6- What were the detection conditions (filters or monochromator…) used to obtain the images in figure 5 top row? The size bar for those figures should be provided.

7- In figure 7, the so-called recognition coefficient is defined in a way that does not match the data displayed in the same table.

Reviewer 3 Report

The authors investigated the theranostic properties of aluminum phthalocyanine nanoparticles (AlPcNP). Also, the fluorescence quenching and the photodynamic properties of AlPcNP were studied. In this work, the authors provided detailed data and careful assignments. This work suggested a valuable strategy to increase the specificity and selectivity of phototheranostic methods using AlPcNP. Therefore, after minor corrections, I recommend this article for publication in this journal.

  1. The quality of the Figures is very low in the PDF version. Thus, the authors should improve the resolution of the figures.
  2. The authors should check the superscript and subscript. (1O2, TiO2, 1O2generation etc.)